# From heterogeneous healthcare data to disease-specific biomarker networks: A hierarchical Bayesian network approach

**Ann-Kristin Becker**[1], **Marcus Dörr**[2,3], **Stephan B. Felix**[2,3], **Fabian Frost**[4], **Hans J. Grabe**[5], **Markus M. Lerch**[4], **Matthias Nauck**[6], **Uwe Völker**[7], **Henry Völzke**[8], **Lars Kaderali**[1] *

**1** Institute of Bioinformatics, University Medicine Greifswald, Greifswald, Germany, **2** Department of Internal Medicine B, University Medicine Greifswald, Greifswald, Germany, **3** German Centre for Cardiovascular Research (DZHK), partner site Greifswald, Greifswald, Germany, **4** Department of Internal Medicine A, University Medicine Greifswald, Greifswald, Germany, **5** Department of Psychiatry, University Medicine Greifswald, Greifswald, Germany, **6** Institute of Clinical Chemistry and Laboratory Medicine, University Medicine Greifswald, Greifswald, Germany, **7** Interfaculty Institute of Genetics and Functional Genomics, Department of Functional Genomics, University Medicine Greifswald, Greifswald, Germany, **8** Institute of Community Medicine, SHIP/KEF, University Medicine Greifswald, Greifswald, Germany

* lars.kaderali@uni-greifswald.de

**Data Availability Statement:** Data from the Study of Health in Pomerania (SHIP) cannot be shared publicly as they contain potentially identifying and

## Abstract

In this work, we introduce an entirely data-driven and automated approach to reveal disease-associated biomarker and risk factor networks from heterogeneous and high-dimensional healthcare data. Our workflow is based on Bayesian networks, which are a popular tool for analyzing the interplay of biomarkers. Usually, data require extensive manual preprocessing and dimension reduction to allow for effective learning of Bayesian networks. For heterogeneous data, this preprocessing is hard to automatize and typically requires domain-specific prior knowledge. We here combine Bayesian network learning with hierarchical variable clustering in order to detect groups of similar features and learn interactions between them entirely automated. We present an optimization algorithm for the adaptive refinement of such group Bayesian networks to account for a specific target variable, like a disease. The combination of Bayesian networks, clustering, and refinement yields low-dimensional but disease-specific interaction networks. These networks provide easily interpretable, yet accurate models of biomarker interdependencies. We test our method extensively on simulated data, as well as on data from the Study of Health in Pomerania (SHIP-TREND), and demonstrate its effectiveness using non-alcoholic fatty liver disease and hypertension as examples. We show that the group network models outperform available biomarker scores, while at the same time, they provide an easily interpretable interaction network.

## Author summary

High-dimensional and heterogeneous healthcare data, such as electronic health records or epidemiological study data, contain much information on yet unknown risk factors that

sensitive medical information on study participants. However, data access can be requested from the Forschungsverbund Community Medicine data access committee (online application form at http://fvcm.med.uni-greifswald.de) for researchers who meet the criteria for access to confidential data.

**Funding:** We acknowledge funding by the German BMBF via the LiSyM grant (FKZ 031L0032). AKB holds an add-on fellowship from the Joachim Herz Stiftung. HJG has received travel grants and speakers honoraria from Fresenius Medical Care, Neuraxpharm, Servier and Janssen Cilag as well as research funding from Fresenius Medical Care. SHIP is part of the Community Medicine Research Network of the University Medicine Greifswald, which is supported by the German Federal State of Mecklenburg- West Pomerania. The funders had no role in study design, data collection and analysis, decision to publish or preparation of the manuscript.

**Competing interests:** The authors have declared that no competing interests exist.

are associated with disease development. The identification of these risk factors may help to improve prevention, diagnosis, and therapy. Bayesian networks are powerful statistical models that can decipher these complex relationships. However, high dimensionality and heterogeneity of data, together with missing values and high feature correlation, make it difficult to automatically learn a good model from data. To facilitate the use of network models, we present a novel, fully automated workflow that combines network learning with hierarchical clustering. The algorithm reveals groups of strongly related features and models the interactions among those groups. It results in simpler network models that are easier to analyze. We introduce a method of adaptive refinement of such models to ensure that disease-relevant parts of the network are modeled in great detail. Our approach makes it easy to learn compact, accurate, and easily interpretable biomarker interaction networks. We test our method extensively on simulated data as well as data from the Study of Health in Pomerania (SHIP-Trend) by learning models of hypertension and non-alcoholic fatty liver disease.

## Introduction

High-throughput technologies and electronic health records allow for digital recording and analysis of large volumes of biomedical and clinical data. These data contain plenty of information about complex biomarker interaction systems, and they offer fascinating prospects for disease research. However, to extract this knowledge from the data and make it accessible, we need models that are accurate, easily interpretable, and compact. Bayesian networks (BNs) are popular and flexible probabilistic models that lie at the intersection of statistics and machine learning and can be used to model complex interaction systems. BNs explicitly describe multivariate interdependencies using a network structure in which the measured features are the nodes and directed edges represent the relationships among those features. Thus, they offer an intuitive graphical representation that visualizes how information propagates. This interpretable structure sets them apart from 'black-box' concepts of other machine-learning methods. Besides, there are well-established algorithms for the automatic learning of Bayesian networks from data, and they are widely used in Systems Biology, e.g., to model cellular networks [1], protein signaling pathways [2], gene regulation networks [3–5], or as medical decision support systems [6]. For a thorough introduction to Bayesian networks see for example Koski and Noble [7] or Koller and Friedman [8].

However, large volumes of biomedical data raise a challenge for computational inference, as in addition to their high dimensionality, other difficulties, such as incompleteness, heterogeneity, variability, strong feature correlation, and high error rates usually co-occur. Considerable manual time and human expertise are therefore necessary to process and format data, including steps of annotation, normalization, discretization, imputation, and feature selection. In addition to the related expenses, these preprocessing steps have a substantial impact on the resulting model [9, 10]. Therefore, we have developed an entirely automated and data-driven workflow that combines Bayesian network learning with hierarchical variable clustering. Our approach tackles many of the mentioned issues simultaneously, while in manual processing, they are usually approached independently. The combination of the two well-established concepts helps to derive precise biomarker interaction models of manageable complexity from unprocessed biomedical data.

Bayesian network learning usually comprises two separate steps: First, the network structure (a directed acyclic graph) is inferred, then, local probability distributions are estimated.

Structure learning can either be carried out using repeated conditional independence tests (constraint-based learning) or search-and-score techniques (score-based learning) [8]. However, as the number of possible network structures grows super-exponentially, available algorithms usually do not scale well to more than 50 to 100 variables. Various heuristic approaches as well as the incorporation of further information, such as sparseness assumptions or more general restrictions of the search space have led to some progress in learning large Bayesian networks [11, 12]. However, due to the complexity of the underlying statistical problem (non-identifiability, non-convexity, non-smoothness), Bayesian network learning from high-dimensional data remains challenging, and often yields inconsistent results. Moreover, the subsequent interpretation of a giant network is just as complex. Because of the mentioned difficulties, published biomedical Bayesian network models are often based on molecular data-sets with homogeneous variables [13–15], as for them, all features can be processed in a similar way. Often, the subsequent analysis concentrates mainly on global network properties. Studies on heterogeneous epidemiological data usually involve smaller models with a preselected set of features, e.g., of cardiovascular risk [16, 17], renal transplantation [18] or liver diseases [19–21].

Because of the way in which biomedical data are gathered, they often contain groups of highly related variables. Some features may be explicitly redundant (like replicated measurements) or multiple features measure the same aspect (like the percentage of body fat and waist circumference). The underlying interaction network (Fig 1A) is then modular or hierarchically modular [22, 23]. This modularity complicates the identification and inference of a Bayesian network even more, as for example many structure learning algorithms penalize for high node degrees that are present in such modules [8]. However, if the modular organization is known, it can be used to simplify the original problem. Instead of aiming for a detailed Bayesian network, a network among groups of similar features can be learned (Fig 1C). Such networks are called *group Bayesian networks*. Group Bayesian networks are smaller and less connected than detailed networks. Moreover, results tend to be more consistent, as the grouping and

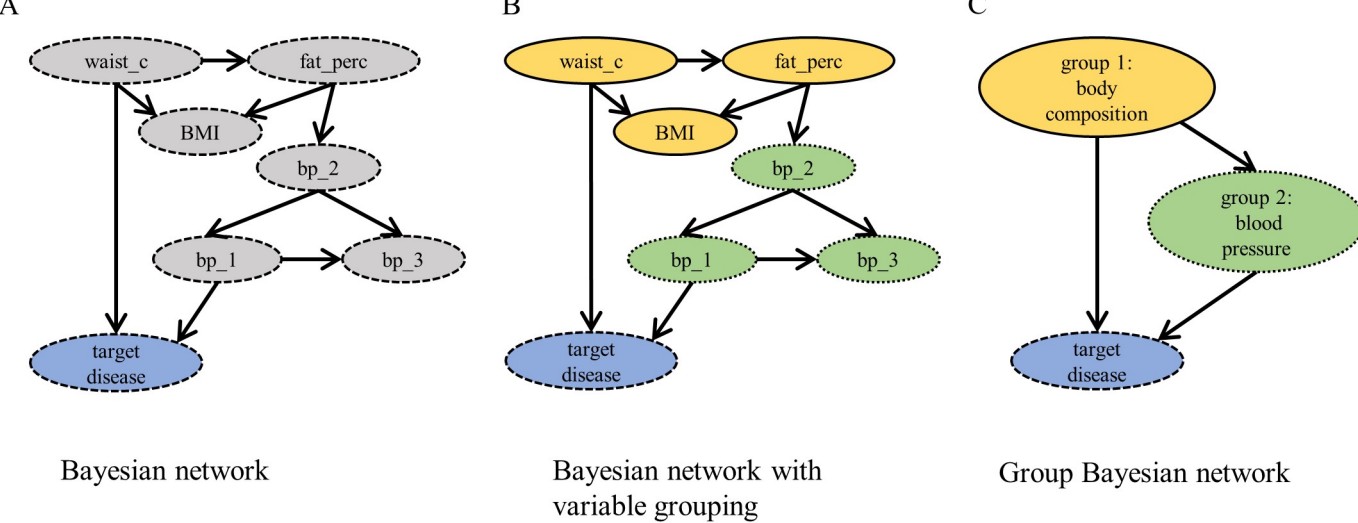

**Fig 1. Hypotetical example Bayesian network with and without variable grouping.** (A) Example model of a modular detailed Bayesian network with variables waist circumference (*waist_c*), body fat percentage (*fat_perc*), BMI and three blood pressure measurements (*blood_pr1, blood_pr2, blood_pr3*) as well as a *target disease*. (B) Possible grouping of the variables in the network. (C) Corresponding group Bayesian network among two groups and the target variable.

aggregation act as denoise filters. Additionally, the abstraction enables the understanding of the larger picture from a system's point of view.

Most publications that have addressed the question of how to learn Bayesian networks of variable groups discuss the problem for a given grouping. This includes the application to pathway or SNP dependencies given detailed genetic data [24, 25]. However, the determination of the number and type of variable groups is a crucial question itself, and it is unlikely that the correct grouping is known for complex and heterogeneous data. On the other hand, there is the concept of *Module Networks*, which is well studied, and algorithms are available to learn modules and their interactions from data [26–28]. But since Module networks were developed in the context of gene regulatory networks, their structural limitations (variables in modules share set of parents and local probability distribution) do not apply to data as we consider here. Likewise, *hierarchical Bayesian networks* (HBNs) [29] define a related, very general concept of tree-like networks, in which leaf nodes represent observed variables and internal layers represent latent variables. HBNs are usually strictly hierarchical. This means that, similar to the architecture of deep neural networks, they restrict all nodes to have parents only in higher layers [30, 31]. Nevertheless, group Bayesian networks can be seen as a special case of loose HBNs. Latent variables in HBNs can theoretically be identified from detailed Bayesian networks, for example, using subgraph partitioning [32]. However, this approach requires the computationally intensive inference of a large, detailed network, and it suffers from the difficulties mentioned above.

We, instead, propose to combine Bayesian networks with hierarchical clustering to learn a grouping of variables as well as the interplay of groups automatically. Hierarchical clustering is one of the most popular methods of unsupervised learning. The output is a dendrogram, which organizes variables in increasingly broad categories. We propose to build group Bayesian networks by aggregating groups learned from hierarchical clustering. As both methods, BNs and clustering, are unsupervised, we enable focusing on a particular target variable of interest—such as a specific disease or condition—during a step of adaptive refinement. We present an optimization algorithm, that, starting from a coarse network, refines important parts of the network downwards along the dendrogram. It zooms automatically into the relevant parts of a network, which are modeled in detail, while other parts stay aggregated. Thus, refined group Bayesian networks offer a good tradeoff between compactness, interpretability, and predictive power.

While some published approaches make use of variable clustering in order to speed up the learning of detailed networks by going from local (within groups) to global (between groups) connections [4, 33, 34], we are not aware of any study addressing the reverse approach.

## Results and discussion

### Algorithm

We here introduce a novel algorithm to significantly simplify the use of Bayesian network models for biomarker discovery (Fig 2). It explicitly integrates a target variable of interest that guides the search through the biomarker network. Our approach exploits the modular structure of large biomedical data and models dependencies among groups of similar variables. To keep the combined search for grouping and network structure feasible, a hierarchical structure acts as a basis for the following network inference procedure. Initially, a dendrogram of the feature space is determined via unsupervised, similarity-based hierarchical clustering. A coarse, preliminary grouping of features is identified, and the data are aggregated in groups using principal components. Then, structure and parameters of a Bayesian network model are fitted. The target variable is kept separated during this procedure so that the resulting model

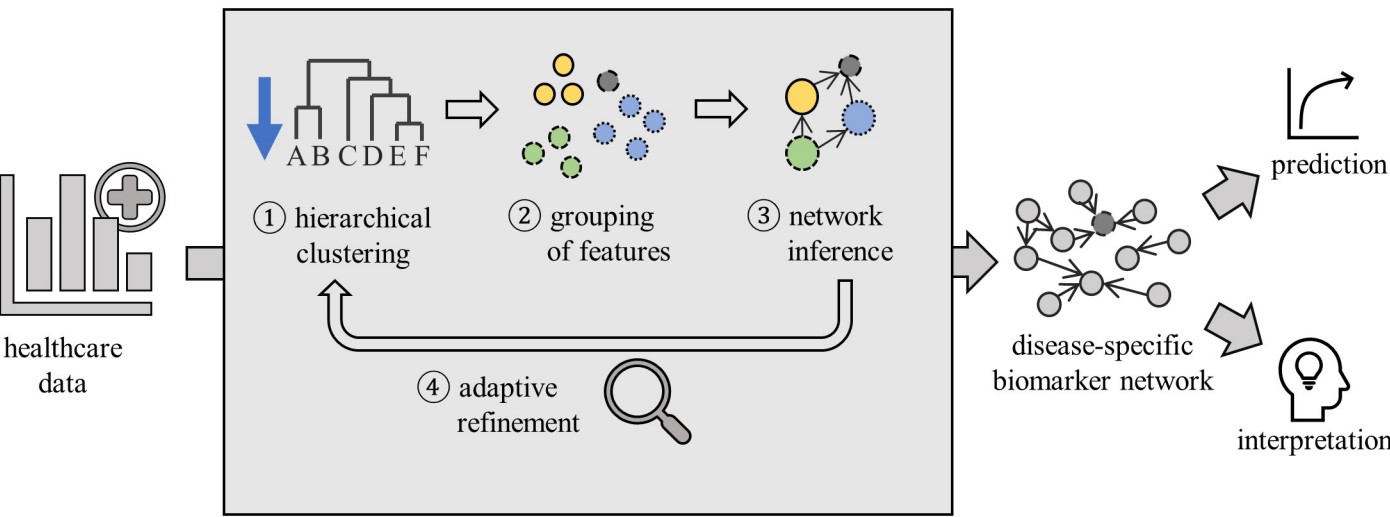

**Fig 2. Schematic outline of the proposed approach to learn group Bayesian networks.** Features of the input data are grouped using hierarchical clustering, then a group Bayesian network is learned. Based on the accuracy of the resulting model, the grouping is refined adaptively downwards along the dendrogram. The output is an interpretable disease-specific biomarker network based on feature groups, which has high predictive accuracy.

can be used for risk prediction and classification. Such groups that were identified to be essential for the prediction of the target variable (i.e., are part of its Markov blanket) are then iteratively refined to smaller clusters. The refinement stops once it no longer helps to improve the predictive performance of the model. We implemented our approach for the construction and refinement of group Bayesian networks using a hill-climbing procedure (Algorithm 1 and 2). The implementation is also available in CRAN from https://CRAN.R-project.org/package=GroupBN [35].

## Evaluating simulated data

We evaluated the proposed approach using simulated data. To generate noisy and heterogeneous data with latent group structure, we randomly created two-layered Bayesian networks (Fig 3A) with one layer of group variables (*layer 1*) and one layer representing noisy and heterogeneous measurements (*layer 0*). Here, the overall aim was to infer the group structure in layer 1 from data in layer 0. For the analysis, we split the algorithm into its three key-tasks, that we evaluated independently: Inference of groups, inference of group network structure, and prediction of a target variable. In the 'standard network inference' approach, the grouping was disregarded for network learning. Instead, a large, detailed Bayesian network was learned, and groups as well as their interactions were only afterwards identified from the network. In the 'group network inference' approach, we contrarily learned the grouping prior to network inference using data-based clustering, as proposed above. For group aggregation, we compared cluster medoids (MED) to first principal components (PC). As a baseline comparison for the quality of the network structure, we additionally inferred the network structure directly from data sampled from layer 1 ('using ground-truth grouping'). We used a partition metric to the ground-truth grouping and the normalized Hamming Distance to the ground-truth network as measures of quality. Lastly, we iteratively chose each variable as target variable and measured the average predictive performance of a detailed network, as well as group networks before and after target-specific refinement. Here, we compared the average prediction error to the applied noise level.

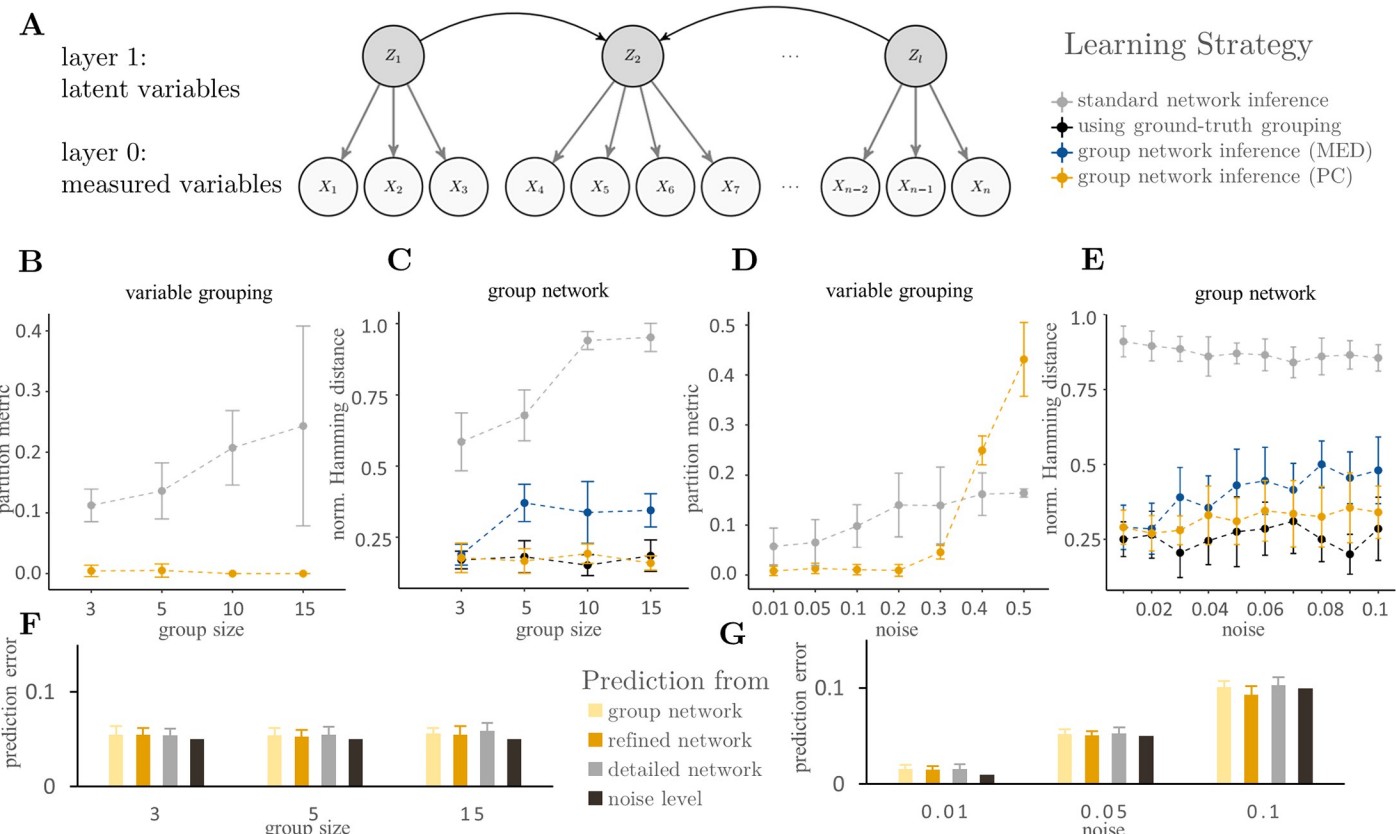

**Fig 3. Results on simulated networks. (A)** The basic model structure used to simulate random networks with latent group structure. Group networks with 20 nodes in layer 1 were learned from simulated data from layer 0 with varying group sizes and noise levels. **(B-C)** Results from the reconstruction of variable grouping and group networks for varying group sizes. y-axes showing partition metric and normalized Hamming distance, respectively. Two types of group network inference—aggregation by principal components (PC) and cluster medoids (MED) – as well a standard network inference approach were used. As a comparison, the ground-truth grouping was used for network inference. **(D-E)** Results from the reconstruction of variable grouping and group networks for varying noise levels. y-axes showing partition metric and normalized Hamming distance, respectively. **(F-G)** Results from the prediction of a target variable for varying group sizes and noise levels, and applied noise level as comparison. y-axes showing the average prediction error.

**Influence of network size and sample size.** We first analyzed the influence of network and sample size on the model quality. The results show that the quality of the network structure is best for high sample sizes and small network sizes (S1 Fig). Overall, the PC-based aggregation is close to the baseline results, followed by the medoid-based aggregation, with the network-based aggregation performing worst. Based on these results, we decided to run the remaining simulations with group networks consisting of 20 nodes at layer 0 and a medium sample size of 500.

**Influence of group size.** Next, we tested the influence of group size on the inference results. We ran simulations with groups ranging from 3 to 15 nodes each. The results show that the identification of variable groups based on a detailed network is impaired with increasing group size. In contrast, data-based clustering enables the detection of the nearly correct grouping independently of the group size. Moreover, even the existence of small groups impedes the inference of the network structure from a detailed network significantly. Especially the number of group connections is underestimated. This effect increases with increasing group size, approaching scores similar to a model without any arcs (Fig 3B). However, data-based clustering enabled the detection of the correct grouping independently of the group size. Aggregation of data before network learning leads to networks that are qualitatively

comparable to networks learned from the group data directly (Fig 3C). Here, the aggregation based on principal components overall achieves better results than with medoids. The prediction of a target variable is mostly positively affected by the grouping (Fig 3F). The refined networks overall perform slightly better when used for predicting the target variable in a cross-validation setting than the detailed models.

**Influence of random noise.**    Finally, we analyzed the influence of random noise. For this purpose, we simulated networks affected by different amounts of random noise in layer 1. The results show that the quality of the inference of groups, as well as group interactions, decreases with increasing noise levels. Data-based clustering outperforms network-based clustering for noise levels up to 35% (Fig 3D). Data aggregation by principal components overall leads to better networks than the use of medoids (Fig 3E). However, a decrease in quality can be noticed for both approaches, as the noise level increases. The average error in prediction of a target variable appears to be in the range of the noise level with slight improvements after target-specific refinement (Fig 3G).

**Discussion of simulation results.**    The simulation results underline, that the aggregation of data increases the quality of the network model compared to group networks that were inferred from detailed networks. This may be explained by the inherent regularization of most structure learning algorithms, that prioritize intra-group interactions in this setting, as those are very strong. Thus, groups tend to be disconnected from each other in a detailed network, even though strong connections are present in the correct network. The proposed combination of hierarchical clustering and network inference puts importance on the inter-group interactions, enabling their accurate inference. Moreover, we observed an overall better performance of aggregation using principal components. This goes along with earlier results on PCA preprocessing for Bayesian networks [10].

## Toy example: Wine data

As a first illustration using a small, real-world example, we demonstrate the capability of the proposed method on benchmark data for clustering of heterogeneous variables. The *wine* dataset [36, 37] contains data on the sensory evaluation of red wines from Val de Loire. Variables contain scorings on origin, odor, taste, and visual appearance of the wines. We study the influence of the wine-producing soil on the properties of the wine.

We examine the difference of 7 wines grown in soil type *Env1* to 7 wines of the class *Reference*, an excellent wine-producing soil. In order to learn the links among the variables, we clustered the data subset (14 samples, 29 variables) hierarchically (Fig 4A). We chose 5 clusters for an initial grouping. Fig 4B and 4C show the group Bayesian network model before and after refinement. Line thickness illustrates the confidence of the learned interaction. The neighborhood of the target variable is modeled more detailed in the refined network (Fig 4C). The network revealed two factors, that mainly distinguish wines from *Soil = Reference* and *Soil = Env1*; namely *Acidity* and *Aroma.quality.before.shaking*. Through these variables, the target is further indirectly linked to two kinds of odor (fruity, flower), as well as a larger cluster comprising measures of odor- and aroma intensity. A closer look at the parameters of the Bayesian network revealed that a wine from the reference soil is typically more fruity, less acidic, and has a higher score in aroma quality and floral aroma.

The arc with the highest confidence was learned among *aroma quality before shaking* and *Soil*. Given a wine with a good aroma quality before shaking, there is an 85% probability according to the model, that this wine is from the reference soil and only 15% that it is from soil class Env1. On the contrary, soil and spiciness or overall balance of a wine are

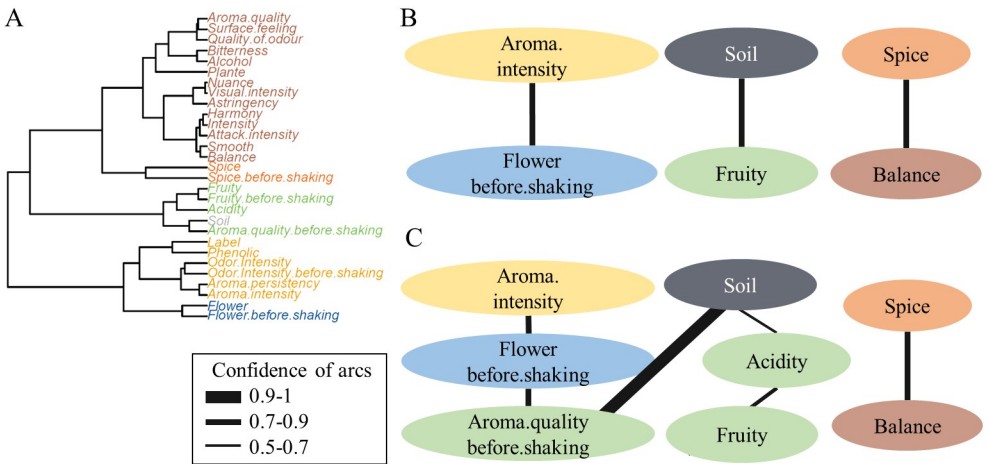

**Fig 4. Toy example: Wine dataset.** (A) Dendrogram of the wine dataset with 5 groups indicated by colour, and the target variable *Soil* separated. (B) Group Bayesian network learned from the wine dataset with 5 groups, colours refer to the grouping. (C) Group Bayesian network after target-specific refinement.

disconnected in the network, indicating that the soil type Env1 does not influence these characteristics significantly.

## Validation with healthcare data: Study of Health in Pomerania (SHIP-TREND)

We further validated the methodology using data from the Study of Health in Pomerania (SHIP-Trend-0) with focus on two common, multifactorial diseases, non-alcoholic fatty liver disease and hypertension. SHIP-Trend is a large-scale cohort study on the general population in Northeast Germany [38]. Interdisciplinary baseline examinations on a total number of 4420 participants were conducted between 2008 and 2012, including a wide variety of assessments. These assessments involve the recording of socioeconomic factors, a detailed questionnaire, measurements of molecular data, preexisting conditions, as well as various clinical tests such as blood counts, imaging techniques, electrocardiography, body impedance analysis and others.

**Application 1: Non-alcoholic fatty liver disease.** Non-alcoholic fatty liver disease (NAFLD) is widely considered a hepatic manifestation of the metabolic syndrome and represents the most common chronic liver disease worldwide, affecting 15-35% of the general population. Hepatic steatosis is the key feature of NAFLD and describes the excessive accumulation of liver fat. Steatosis is diagnosed if the amount of intrahepatic triglycerides exceeds 5% [39]. Simple hepatic steatosis may progress to non-alcoholic steatohepatitis (NASH), marking the most crucial step in the development of severe liver dysfunction with poor prognosis. Causes of the disease, as well as its progression, are still poorly understood. Today, liver biopsy is the gold standard to diagnose NAFLD [40] and its stage. However, besides its sampling bias, liver biopsy always involves risk of complications. Apart from that, imaging techniques like ultrascan or magnetic resonance imaging are used. The development of cheaper and reliable noninvasive techniques to diagnose NAFLD are of urgent need—also with regard to prevention. Therefore, several biomarker scores have been proposed in the last years, including the Fatty Liver index [41], Hepatic Steatosis Index [42, 43], and NAFLD ridge score [44], all of which combine 3 to 6 different anthropometric parameters and biochemical tests. They allow for a cheap and noninvasive screening for steatosis in the general population. On their respective

**Table 1. Prediction results of NAFLD models.**

| Model | AUROC | ± sd | AUPRC | ± sd |
|---|---|---|---|---|
| Hepatic Steatosis Index | 0.68 | ±0.04 | 0.24 | ±0.04 |
| Fatty Liver Index | 0.78 | ±0.05 | 0.34 | ±0.05 |
| NAFLD ridge score | 0.73 | ±0.05 | 0.29 | ±0.04 |
| logistic regression | 0.78 | ±0.03 | 0.37 | ±0.05 |
| detailed Bayesian network | 0.74 | ±0.02 | 0.31 | ±0.05 |
| group Bayesian network | 0.79 | ±0.04 | 0.35 | ±0.05 |
| refined group Bayesian network | 0.82 | ±0.03 | 0.42 | ±0.04 |

Evaluation of available steatosis scores, logistic regression and different Bayesian network models on SHIP Trend data in terms of discrimination. The table shows area under receiver-operator curve (AUROC), and area under precision-recall curve (AUPRC) under 10-fold cross validation (mean and standard deviation). Predictions from Bayesian network models were obtained using likelihood weighting by taking all nodes but the target as evidence. Best scoring steatosis biomarker score and best scoring Bayesian network model are highlighted.

original datasets, these scores achieved an area under the receiver-operator curve (AUROC) between 0.81 and 0.87, thus leaving a substantial proportion of false positive and false negative results. On the SHIP Trend data, the AUROC lies significantly lower, between 0.67 and 0.78 (Table 1). The area under the precision-recall curve (AUPRC), which has its focus on the underrepresented class of positive cases, ranges from 0.24 to 0.34.

We applied the proposed group network approach to the SHIP-Trend data. Compared to a detailed network, the aggregation of data into groups already improved the prediction of steatosis in a Bayesian network (Table 1). The unrefined group network achieved an AUROC score of 0.79 in a cross validation setting. The score is comparable to the one reached by logistic regression and the FLI, which we found to be the best performing biomarker score on the SHIP Trend data of the three tested ones. The refinement procedure resulted in an improved final AUROC score of 0.82 and an AUPRC of 0.42.

We then fit a final model on the complete dataset for interpretation. Hierarchical clustering of the data revealed 17 groups of features. The final network model (Fig 5 and S4 Fig) has an average neighbourhood size of 2.5, an average group size of 16 and also achieved an AUROC of 0.82. Fig 5A shows the complete network structure, in which sex and age are both hubs. Fig 5B shows only the target variable and its surrounding. The group names have been chosen

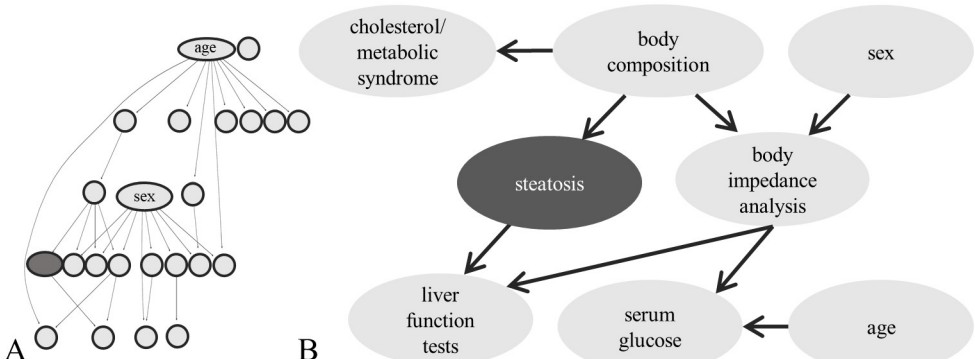

**Fig 5. Steatosis network model.** (A) Structure of the complete, refined group Bayesian network model for hepatic steatosis. (B) Extract from the group network including the target variable *steatosis*, its Markov blanket and surrounding.

manually according to the included variables. The detailed grouping can be found in S1 Data. Steatosis has one parent node, which is a group of variables related to body composition, including body mass index, waist circumference, body fat and others. This group is further linked to a group including cholesterol and triglyceride levels, as well as a group including raw results of the body impedance analysis (BIA). The child node of the target comprises different variables related to serum liver function tests (alanine aminotransferase, aspartate transaminase, gamma-glutamyl transferase). Sex and serum glucose levels are indirectly linked to the group of liver function tests via BIA results.

We further evaluated the distance of the features to the target in the network. In the moralized network, the average distance to the target variable is 2.09. The predictors that have been used in the liver scores are closer than average, with in average only 1.5 arcs distance to the target. For the the FLI, three out of the used four predictors (BMI, waist circumference, triglycerides and GGT) are within the Markov blanket (mean distance 1.25). This overlap might explain the similarity in prediction performance. It shows that meaningful features have been learned by the network. Moreover, the network illustrates clearly the strong relation of steatosis with obesity and the metabolic syndrome. However, different from pure prediction scores, the interpretability of the proposed model enables the understanding of how and why a prediction is made, and, by this, it shows also what may be overlooked. According to the model, and consistent with earlier studies, around 10% of steatosis cases do not go along with multi-organ metabolic abnormalities and obesity [45, 46]. These cases stay hardly detectable without imaging techniques.

**Application 2: Hypertension.** As a second example, we analyzed the SHIP-Trend data with a focus on hypertension. Hypertension describes the condition of persistently elevated blood pressure in arteries and is a major risk factor for coronary artery disease, stroke, heart failure, and overall end-organ damage (heart, kidneys, brain, and eyes). Blood pressure measurements monitor systolic (contraction) and diastolic (relaxation) pressures. Hypertension is typically diagnosed if the systolic pressure exceeds 140 mmHg or the diastolic pressure exceeds 90 mmHg. It is known to have a substantial heritability (estimated in the range of 30–55%) [47]. Moreover, many risk factors of hypertension are well established, including obesity, age, stress, or chronic conditions, such as diabetes or sleep apnea.

For our analysis, incident hypertension was defined as blood pressure above 140/90 mmHg or self-reported antihypertensive therapy. The target variable was not well connected within a detailed network learned from SHIP-Trend data, which is why a mean AUROC of only 0.55 is achieved in a cross validation setting for training as well as test sets. The refined group network model, however, achieves an AUROC score of 0.84 and an AUPRC of 0.81 (Table 2), which is comparable to other recent hypertension risk-prediction models and results on an earlier SHIP cohort [48, 49].

**Table 2. Prediction results of hypertension models.**

| Model | AUROC | ± sd | AUPRC | ± sd |
|---|---|---|---|---|
| logistic regression | 0.82 | ±0.02 | 0.78 | ±0.03 |
| detailed Bayesian network | 0.55 | ±0.04 | 0.57 | ±0.06 |
| group Bayesian network | 0.80 | ±0.02 | 0.76 | ±0.04 |
| refined group Bayesian network | 0.84 | ±0.03 | 0.81 | ±0.02 |

Evaluation of logistic regression and different Bayesian network models on SHIP Trend data for the prediction of hypertension. The table shows area under receiver-operator curve (AUROC), and area under precision-recall curve (AUPRC) under 10-fold cross validation (mean and standard deviation). Predictions from Bayesian network models were obtained using likelihood weighting by taking all nodes but the target as evidence. Best scoring Bayesian network model is highlighted.

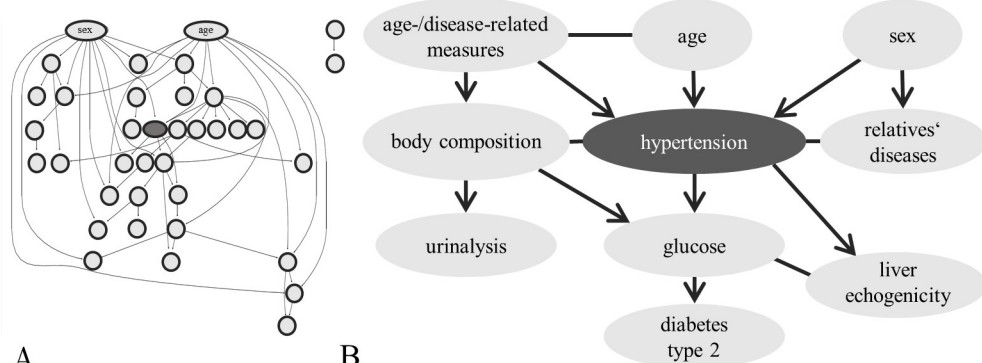

**Fig 6. Hypertension network model.** (A) Structure of the complete, refined group Bayesian network model for hypertension. (B) Extract from the group network including the target variable *hypertension*, its Markov blanket and surrounding.

The final group Bayesian network from 28 groups, as determined by the aggregation levels, is densely connected. After refinement, the network (Fig 6 and S3 Fig) has an average neighbourhood size of 3.5 and an average group size of 9.4. The target variable has three parents in the network, which are age, sex, and a cluster of more general age- and disease-related measures (including the number of doctoral visits, and information on employment/retirement). Further, a cluster of diseases of first degree relatives (including hypertension, heart attack, stroke, diabetes) and a cluster of measures of body composition are directly attached to the target. A group around fasting glucose level as well as a group around liver echogenicity are children of the target variable in the network. Via body composition, hypertension is further linked to a group of urinalysis results, as they show frequent consequences of hypertensive kidney injury. The network clearly visualizes the heritability of hypertension, as well as promoting environmental factors. The detailed grouping can be found in S2 Data.

## Conclusion

Bayesian networks provide a powerful and intuitive tool for the analysis of the interplay of variables. In this work, we introduced a novel algorithm to infer Bayesian biomarker and risk factor networks from heterogeneous and high-dimensional healthcare data. Our approach combines Bayesian network learning and hierarchical variable clustering. By this means, it supersedes many of the usually necessary manual preprocessing steps and reduces the complexity of the computations, while preserving model interpretability. We introduced an optimization algorithm for adaptive network refinement, which emphasizes a variable of interest and enables the automated refinement leading to small yet precise disease-specific models. The results on simulated data, test data and real-world epidemiological data verify the ability of the approach to successfully reveal important biomarker and risk factor interactions. Moreover, we showed that the increased interpretability of the model does not restrain its predictive performance, which was in both biomedical examples equal or better than well-established purely predictive models. Our method is suitable for an in-depth analysis of biomarker systems, but apart from this, it can also be used as a quick summary and visualization tool for large data prior to further evaluation. Our findings add to a growing body of literature on the use of machine learning and artificial intelligence in medicine, and they facilitate multivariate data analysis, visualization, and interpretation.

The purpose of this study was to investigate how hierarchical variable grouping and Bayesian network learning can be combined to overcome the limitations of network inference on high-dimensional and heterogeneous data. The proposed methodology provides the framework to effectively learn Bayesian networks of manageable complexity without manual steps of feature selection. Our method could be applied to all types of tabular data with many features and high enough sample size for which the interest lies mainly in feature interactions. A crucial step in our procedure is the aggregation of groups for network learning. We found that in the studied data sets, groups of variables were often reflecting highly similar information. The use of single principal components as cluster representatives was therefore mostly sufficient and yielded reasonable clusters. However, depending on the complexity and the aim of the application, the results may be improved by the use of more sophisticated and more accurate aggregations, for example using multidimensional cluster representatives. However, higher precision in the modeling of variable groups would, in turn, significantly increase the computation time and complicate the interpretation. Note also, that data have to satisfy additional assumptions in order to be exactly modeled as group networks with two- or more-dimensional nodes, as studied by Parviainen and Kaski [25]. The same applies to a more complex grouping that allows overlapping clusters. For future studies, in particular a dynamic generalization of the approach using dynamic Bayesian networks is planned to enable the use of longitudinal study data for prognosis. We plan to also include molecular data in order to allow the integrative analysis of multi-omics and epidemiological data. By this, the proposed methodology offers the possibility to reveal yet unknown biomarker and risk factor relations, and to gain new insight into molecular disease mechanisms.

## Methods

### Group Bayesian networks and adaptive refinement

We implemented an approach to learn group Bayesian networks (Algorithm 1). Prior to the procedure, a hierarchy of the feature space has to be determined by hierarchical clustering. An initial variable grouping is determined by cutting the dendrogram into $k$ clusters and cluster representatives are calculated as first principal components. A target variable can be chosen, which is kept separated. Then, a Bayesian network structure is learned using the discretized version of the cluster representatives and parameters are fitted.

Moreover, we implemented a refinement algorithm for such group Bayesian networks via a divisive hill-climbing approach (Algorithm 2). The current network model is used to predict class probabilities of the target variable using all remaining nodes as evidence, and a prediction score is calculated. As usual for hill-climbing approaches, in each step, all neighbouring states of the current model are evaluated. Those include all models, in which one group was split into two smaller groups along the dendrogram. From the neighbouring states, the model with the highest score improvement is chosen. The procedure is repeated until no further improvement is possible. Random restarts and perturbations are possible to escape from local optima.

To reduce the computation time, the tested splits may be restricted to the Markov blanket of the target variable or a certain maximal distance in the current network. This requires the initial grouping to be detailed enough, so that all important direct relations could be learned. The plot of the aggregation levels for different cluster numbers may help to choose an initial number of clusters that gives a good trade off between data compression and information loss.

If useful, further features besides the target can be chosen to be separated from their groups as well, as for example sex or age that are well-known confounders in many problems.

**Objective function.**    Throughout the refinement, we use the the cross-entropy as objective function for a binary outcome, also known as log-loss, weighted by the class proportions. It

can be calculated as

$$H(o, p) = -\sum_{i=1}^{N} w_i o_i \log (p_i),$$

where $o \in \{0, 1\}^N$ is the vector of observations, $p \in [0, 1]^N$ is the vector of predicted class probabilities, and $w \in (0, 1)^N$ is the vector of weights with $w_i = \frac{\#\{o=o_i\}}{N}$. Using the class proportions as weights ensures that both outcome classes have an equal share in the total score, independent of their proportion in the training data. The adjustment is important, as often the target variable is heavily imbalanced. Without these weights, the optimization prioritizes models that primarily predict the majority class, as those have high accuracy. In case of a continuous target variable the objective function must be respectively altered.

To account for the stochasticity of the probability estimates $p_i$, which are based on likelihood-sampling, we estimate an uncertainty range of $H(o, p)$ over 20 runs and accept a more complex model only if its score exceeds this range.

**Algorithm 1**: Group Bayesian network

```
1: procedure GROUPBN(D, g, t)
2:                         // D: dataset, g: feature grouping
3:                         // t: name of target variable
4:
5:    D_g ← AGGREGATE(D, g)       //aggregate data in groups g
6:    D_{g,t} ← SEPARATE(D_g, t)    //separate t from its cluster
7:    S ← BNSL(D_{g,t})          //structure learning
8:    P ← BNPL(D_{g,t}, S)        //parameter learning
9:    M ← (S, P)
10:
11:   return M               //return group BN model
12: end procedure
```

**Algorithm 2** Adaptive Refinement

```
1: procedure GROUPBN_REFINEMENT(D, H, k, t)
2:                         //D: dataset, H: feature hierarchy
3:                         //k: initial number of groups
4:                         //t: name of target variable
5:
6:    g ← CUT(H, k)              //cut the hierarchy into k groups
7:    M← GROUPBN(D, g, t)        //learn inital group network
8:    c← LOSS(M, t)             //calculate loss function for target
9:
10:   repeat                  //refinement step
11:      B← MARKOVBLANKET(M)        //set of splits to be tested
12:
13:      for b in B do            //Evaluate all neighbouring models
14:         g_b ← SPLIT(H, g, b)       //split cluster b according to H
15:         M_b ← GROUPBN(D, g_b, t)    //and learn new model
16:         c_b ← LOSS(M_b, t)
17:      end for
18:
19:      if min c_b < c then        //if improvement is possible
20:         b* ← which.min(c_b)
21:         g ← g_{b*}
22:         M ← M_{b*}              //Replace M with best model
23:         c ← c_{b*}
24:      else break
25:
26:   end repeat
```

```
27:
28:    return M                //return refined group BN model
29: end procedure
```

## Hierarchical clustering and data aggregation

To identify groups of similar variables, an agglomerative similarity-based hierarchical variable clustering is used. As the method needs to be applicable to high-dimensional and heterogeneous data (qualitative and quantitative variables), we used the algorithm implemented in the ClustOfVar package in R [50]. A key step of the clustering is the determination of a synthetic central variable for each cluster, which is calculated as the first principal component from the PCAmix method [51]. PCAmix combines principal component analysis and multiple correspondence analysis. For this procedure, the data matrices are internally standardized, concatenated, and factorized respectively. The homogeneity of a cluster is calculated as the distance of all cluster variables and its representative. This distance is based on squared correlation and correlation coefficient.

## Bayesian networks

A *Bayesian network* (BN) is a pair $(G, \Theta)$, where $G$ is the structure that represents a random vector $X = (X_1, \ldots, X_n)$ and its conditional dependencies via a directed acyclic graph. $\Theta$ is the set of parameters. The parameter set $\Theta$ consists of the local conditional probabilities of each node $X_i$ given its parents in the graph. Throughout this section, we denote the set of parents of a node $X_i$ by $\Pi(X_i)$. The parameters are of the form

$$\theta_i = \mathbb{P}(X_i \,|\, \Pi(X_i)).$$

In case of discrete random variables they are conditional probability tables. A Bayesian network encodes the local Markov property, that is, each variable $X_i$ is independent of its nondescendants conditioned on its parents. A general factorization of the joint probability distribution of $X_1, \ldots, X_n$ is given by

$$\mathbb{P}(X_1, \ldots, X_n) = \prod_{i=1}^{n} \mathbb{P}(X_i \,|\, X_{\Pi(i)}) = \prod_{i=1}^{n} \theta_i$$

accordingly. The *Markov blanket* of a node contains its children, its parents and its children's parents. It can be shown, that given the nodes in the Markov blanket, a node is conditionally independent of all other nodes in the network. It, thus, contains all the nodes that are most important for predicting the node itself. The *moralized* counterpart of a Bayesian network is an undirected graph in which each node is connected to its full Markov blanket. It can be constructed by adding arcs between all nodes that have a common child and are not directly connected.

**Data discretization.** The majority of the available BN structure learning algorithms assume that all variables in a Bayesian network are discrete. Hybrid approaches that can handle a mixture of discrete and continuous features include parametric models (i.e., Conditional Linear Gaussian Networks), with the drawback that they restrict the type of distribution and the structure space. More complex nonparametric approaches (see for example Schmidt et al. [52]) are computationally demanding and do not scale well to high-dimensional data. As an alternative, continuous features may be discretized. This simplifies the interpretation and enables the use of well-established algorithms for discrete Bayesian networks. Thus, we decided to discretize the cluster representatives prior to structure learning. Note that for clustering itself, the unprocessed data are used. As the cluster representatives are often multimodal,

we use an unsupervised, density-approximative discretization approach. First, significant peaks in the estimated probability density function of a variable are determined. These peaks are then used to initialize a one-dimensional k-means clustering. This procedure allows the binning, and the number of bins itself, to be directly estimated from the data. If only one significant peak is present, distribution quartiles are used for binning.

**Equivalence classes of Bayesian networks (CPDAGs).**   As several graph structures encode the same conditional independence statements (*Markov Equivalence*), they cannot be distinguished based on observational data alone. As usual, we use completed partially directed acyclic graphs (CPDAG) to represent the inferred equivalence class. In a CPDAG, arcs with undetermined direction are drawn as undirected arcs.

**Bayesian network structure learning.**   Our general approach, does not depend on a specific structure learning algorithm, but works with every available one. For the reported applications, we used the score-based hill-climbing algorithm, as implemented in the bnlearn package [53]. The BIC was chosen as the target function, as it is locally and asymptotically consistent and does not include any hyperparameters. The BIC of a model structure $G$ is defined as

$$\mathrm{BIC}\left(G \,|\, \mathcal{D}\right) \coloneqq \log \mathbb{P}(\mathcal{D} \,|\, G) + \frac{d}{2} \log\left(N\right),$$

where $d$ is the model dimension (the number of free parameters) and $N$ is the number of observations. The BIC is asymptotically and locally consistent and decomposes to parts that are only dependent on one variable $X_i$ and its parents $\Pi(X_i)$. For categorical random variables $X_1, \ldots, X_n$, these parts can be calculated as

$$\mathrm{BIC}\left(X_i, \Pi(X_i) \,|\, \mathcal{D}\right) \coloneqq - \sum_{j}\sum_{k} N_{ijk} \log \frac{N_{ijk}}{\sum_j N_{ijk}} - \frac{q_i(r_i - 1)}{2} \log\left(N\right), \qquad (5)$$

where $N_{ijk}$ is the number of observations in which $X_i = k$ and $\Pi_G(X_i) = j$, $q_i$ is the number of possible states of the parents $\Pi_G(X_i)$ and $r_i$ the number of possible states of $X_i$ itself.

Throughout the adaptive refinement steps, the hill-climbing procedure was initialized with the current network structure and the two new groups, formed by splitting, were embedded into this structure. To escape from local optima, 10 restarts were performed in each run with a number of perturbations depending on the total network size (10% of current number of arcs, at least 1).

Structure learning was repeated 200 times using nonparametric bootstrapping to reduce the number of false positive arcs and add only arcs with high confidence to the model (*model averaging*). The confidence threshold for inclusion of an arc was determined using adaptive thresholding, as suggested in [54].

**Bayesian network parameter learning.**   A Bayesian parameter estimation was performed using the previously determined structure. We used a uniform prior and an imaginary sample size of 1.

## Simulating networks

To generate noisy and heterogeneous data with latent group structure, we sampled two-layered Bayesian networks (Fig 3A) with a layer of (latent) group variables (layer 1), as well as a layer of noisy child variables, reflecting the information of the group variables plus measurement noise (layer 0). Arcs among group variables were sampled using Melancon's and Philippe's Uniform Random Acyclic Digraphs algorithm, which generates graphs with a uniform probability distribution over the set of all directed acyclic graphs. Child nodes were then connected to every group node. We parameterized the group variables using a randomly chosen Dirichlet

distribution, whereas the child nodes could have both, a continuous or discrete range, to simulate heterogeneity. Random noise was introduced via the parameters. For continuous features, a Gaussian noise was added; for discrete features, the distribution was respectively altered. We used these network models to simulate random samples from the joint distribution using forward sampling. By this, several simulated datasets could be created based on the same network model. They were used to assess the quality of the different approaches of group network inference under varying group size, noise level, sample size and network size. Data sampling and network learning were repeated 100 times for each scenario. In the standard network inference approach, the grouping was disregarded for network learning. Instead, a detailed network structure was learned among all variables in layer 0, which was afterwards used to identify groups and group network structure. For identification of the groups, hierarchical community detection was used. The resulting dendrogram was cut at each level, and the grouping that was closest to the true grouping in terms of the evaluation metric was chosen. To aggregate the detailed network into a group network, the ground-truth grouping was applied. As arcs between variables of different groups were only rarely learned, an arc was added to the group network, whenever at least one arc between any two variables from two groups was present. For the group network inference approach, the respective steps of the proposed algorithm were applied.

## Evaluation metrics

**Partition metric.** To compare different variable groupings, we used an entropy-based partition metric [55]. It is zero, if two groupings are identical, and returns a positive value otherwise.

**Structural hamming distance (SHD).** To compare learned Bayesian network structures to the true latent structure, we used the Structural Hamming Distance (SHD). The SHD of two CPDAGs is defined as the number of changes that have to be made to a CPDAG to turn it into the one that it is being compared to. It can be calculated as the sum of all false positive, false negative and wrongly directed arcs. In order to evaluate the quality of inferred group networks, we calculated the SHD of the inferred network and the ground-truth model, and normalized it to the number of arcs within the ground-truth model.

**Area under the curve.** To evaluate the discriminative performance of a model, we compared the area under the receiver-operator (AUROC) as well as the precision-recall curve (AUPRC) in a 10-fold cross validation setting. We calculated the metrics using the PRROC package [56, 57].

## SHIP-trend data preprocessing

The initial set of features was the same for both SHIP Trend examples. As a first step, the set of participants was reduced to those, for which the related diagnosis was present. Further steps included the removement of context-specific variables and features with high amounts of missing values.

**NAFLD.** As target variable for the NAFLD-specific analysis of the SHIP Trend data, we chose the presence of hepatic steatosis diagnosed based on liver MRI. An MRI of the liver was conducted and evaluated for a subset of 2463 participants of the cohort. Probands with a significant intake of alcohol (more than 20 g/day in women, more than 30 g/day in men based on the last 30 days) were excluded from the analysis. Features related to sonography of the liver or earlier diagnoses of steatosis were removed, too (n = 14). From the original dataset, we further removed features that contained more than 20% of missing values (n = 59, S4 Fig). The threshold was chosen to remove measurements that were done for specific patient subgroups only

**Table 3. Processing times.**

|                                  | NAFLD   | Hypertension |
|----------------------------------|---------|--------------|
| number of features               | 407     | 328          |
| number of probands               | 2311    | 4403         |
| hierarchical clustering          | 9m 55s  | 13m 26s      |
| initial group BN                 | 1m 08s  | 2m 57s       |
| group BN refinement (per iteration) | 2m 34s | 5m 11s     |

Individual processing times are stated for initial hierarchical clustering, learning of an initial group BN, and the average time needed for one refinement iteration. It must be noted that processing times depend highly on the chosen structure learning algorithm, the number of groups and the number of neighboured models.

(like, e.g., hormone measurements, differential haematology). Our final dataset comprises 2311 participants and 407 features. The prevalence of NAFLD is 18%.

**Hypertension.**  In SHIP Trend, blood pressure of each proband was measured three times. The average pressure of the latter two measurements was used for diagnosis of hypertension. Probands were classified as hypertensive if their measured systolic pressure exceeded 140 mmHg or the diastolic pressure exceeded 90 mmHg or they reported to receive antihypertensive treatment. Our hypertension model is based on data of 4403 participants (2123 cases of hypertension). From the original dataset we excluded features, that had more than 20% of missing values (n = 63, S4 Fig). We removed all features that contain further information on the blood pressure and earlier diagnoses or treatment of hypertension (n = 35), as well as 54 features related to medication that was related to treatment of hypertension or had extremely low variance (e.g., multiple forms of beta blockers).

## Cross-validation

For comparison of the predictive power of different liver scores, logistic regression and Bayesian network models, we split the data into 10 folds. The liver scores did not have to be trained and were applied to all 10 folds separately to obtain mean and standard deviation. Bayesian network models were trained ten times on 9 of 10 folds and tested on the remaining fold, as usual. As comparison, a regularized logistic regression model was trained and tested. The same folds were used for all tests

## Computations and code availability

All computations were performed using R version 3.6.2 [58] on a Unix workstation with 16 GB RAM and an eight-core Xeon E5-1620 v3 processor running Ubuntu 16.04.6. An implementation of Algorithms 1 and 2 is available on CRAN [35]. Processing times for Hypertension and NAFLD-models are given in Table 3.

## Supporting information

**S1 Fig. Simulation results: Influence of sample size and network size.** Results of the reconstruction of group networks for varying sample sizes. **A** Group networks with 5 nodes.**B** Group networks with 20 nodes. On the basis of these simulations, we decided to run the remaining simulations with group networks of size 20 and a medium sample size of 500. (TIF)

**S2 Fig. Steatosis network.** Group Bayesian network with target variable steatosis.
(TIF)

**S3 Fig. Hypertension network.** Group Bayesian network with target variable hypertension.
(TIF)

**S4 Fig. Missing values in SHIP Trend data.** Histograms of missing values in % per variable **A**
for the subset of participants included in steatosis model **B** for the subset of participants
included in hypertension model.
(TIF)

**S1 Data. Steatosis grouping.** Grouping of steatosis network. Features are sorted by their centrality in the cluster.
(CSV)

**S2 Data. Hypertension grouping.** Grouping of hypertension network. Features are sorted by
their centrality in the cluster.
(CSV)

## Author Contributions

**Conceptualization:** Ann-Kristin Becker, Lars Kaderali.

**Data curation:** Ann-Kristin Becker, Marcus Dörr, Stephan B. Felix, Fabian Frost, Hans J.
Grabe, Markus M. Lerch, Matthias Nauck, Uwe Völker, Henry Völzke.

**Formal analysis:** Ann-Kristin Becker.

**Funding acquisition:** Lars Kaderali.

**Investigation:** Ann-Kristin Becker.

**Methodology:** Ann-Kristin Becker, Lars Kaderali.

**Project administration:** Lars Kaderali.

**Resources:** Marcus Dörr, Stephan B. Felix, Fabian Frost, Hans J. Grabe, Markus M. Lerch,
Matthias Nauck, Uwe Völker, Henry Völzke, Lars Kaderali.

**Software:** Ann-Kristin Becker.

**Supervision:** Lars Kaderali.

**Validation:** Ann-Kristin Becker.

**Visualization:** Ann-Kristin Becker.

**Writing – original draft:** Ann-Kristin Becker.

**Writing – review & editing:** Marcus Dörr, Stephan B. Felix, Fabian Frost, Hans J. Grabe, Markus M. Lerch, Matthias Nauck, Uwe Völker, Henry Völzke, Lars Kaderali.

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
