## [Decision Letter · Decision Letter 0]

12 Nov 2020

Dear Prof. Dr. Kaderali,

Thank you very much for submitting your manuscript "From heterogeneous healthcare data to disease-specific biomarker networks: a hierarchical Bayesian network approach" for consideration at PLOS Computational Biology. As with all papers reviewed by the journal, your manuscript was reviewed by members of the editorial board and by several independent reviewers. The reviewers appreciated the attention to an important topic. Based on the reviews, we are likely to accept this manuscript for publication, providing that you modify the manuscript according to the review recommendations.

Sincerely,

Alison Marsden

Associate Editor

PLOS Computational Biology

Florian Markowetz

Deputy Editor

PLOS Computational Biology

[LINK]

Reviewer's Responses to Questions

**Comments to the Authors:**

Reviewer #1: TITLE

From heterogeneous healthcare data to disease-specific biomarker networks: a hierarchical Bayesian network approach

AUTHORS

A-K. Becker, M. Dorr, S.B. Felix, F. Frost, H.J. Grabe, M.M. Lerch, M. Nauck, U. Volker, H. Volzke, L. Kaderali

DESCRIPTION

The paper proposes an automatic algorithm that hierarchically refines the structure of a Bayesian network to detect groups of homogeneous features and to learn their conditional relation. After an initial presentation of the algorithm, the authors compare the performance of their approach on a synthetic dataset generated from a parametric family of networks, and assess the ability of the proposed approach to recover the original Bayesian network topology and parameters. Their approach is then compared with other strategies to handle groups in Bayesian networks and by aggregation strategies using medoids and first principal components. The proposed approach is tested on a toy model which is used to determine factor distinguishing wines produced from two different types of soils. This is followed by an analysis of a large collection of electronic health records, focusing on two conditions whose early diagnosis is critical for positive long term outcomes, i.e., non-alcoholic fatty liver disease and systemic hypertension. Refined group Bayesian networks show superior performance than commonly adopted clinical indices, logistic regression and Bayesian networks with different group handling strategy.

The paper is interesting and well written. Publication is recommended, I have only a few minor comments for the authors to address.

THINGS THAT ARE NOT CLEAR OR SHOULD BE EXPLAINED FURTHER

- pag. 5 section "Evaluating simulated data". While the difference between approaches to aggregate groups (i.e., medoids and first principal components) is clear, the exact meaning of the other methods such as "network-based" and "using group data" in Figure 3 is not clear. Is "network-based" an approach where group information is disregarded and "group Bayesian network" an approach where the group separation is initially a-priori enforced and left unaltered? The authors should further explain the nature of the approaches they are comparing against, adding appropriate citations from the literature, if appropriate.

- In my opinion, the policy of the journal where the result section precedes the method section is suboptimal for this paper. Many questions that arise in terms of precisely defining how the numerical tests are performed on both synthetic and real datasets find answers in the method section. I therefore think the paper would benefit from switching the result and method sections.

- I was wondering if the authors could add some more detail on how exactly the predictions from the Bayesian network model were obtained in Table 1. Were all other variables assigned as evidence and the most likely value of the variable "steatosis" inferred using a max-product algorithm? Or just variables belonging to the Markov blanket of the variable "steatosis" were used, independent on observations on other variables being missing?

- It is not clear why a table like Table 1 is not provided for the application on systemic hypertension.

- The authors should report the processing times required for the two real dataset analyzed in the papers, i.e., 2311-407 and 4403-328 participant-features for the NAFLD and hypertension datasets, respectively. Specifically, they should report the time required to perform the initial hierarchical clustering, structure learning with group refinement, parameter learning and prediction.

SYNTAX, ETC.

TITLE, ABSTRACT, REFERENCES

- Title and abstract appear to be appropriate.

- Please review reference 15.

RECOMMENDATION

Publication of the paper is recommended.

Reviewer #2: This study presents an approach to analyze medical data using Baysian networks with hierarchical clustering. The authors provide example applications of this model first to a “toy” model and then to hypertension and non-alcoholic fatty liver disease (NAFLD) using a database of clinical data.

I have the following comments/concerns/questions about this manuscript:

The AUROC for the unrefined detailed Bayesian network was significantly lower for hypertension than for NAFLD. Why are these thought to be so different?

For the SHIP Trend data used, it would be useful if the authors provided a little more detail on the input variables and indicate the total number of variables. Were these the same for both the hypertension and NAFLD analyses?

For the SHIP Trend data used, how common were missing variables in the data set used for the analyses? The manuscript indicates that for the both clinical analyses variables with greater than 20% missing data were excluded. Do the authors have any estimate of the effect that missing variables had on their results?

The results for the NAFLD analyses were compared with NAFLD clinical risk prediction models. Why wasn’t the same done with the hypertension analyses? A number of such models have been reported.

Minor:

The clinical standard is to diagnose hypertension when the blood pressure is elevated on repeated measurements rather than a single measurement. Was that the case for the subset of patients diagnosed as being hypertensive based on blood pressure readings?

AUROC and AUPRC are only defined in a figure caption, not in the text.

In Figure 5, BIA is not defined.

**Have all data underlying the figures and results presented in the manuscript been provided?**

Reviewer #1: **No: **The dataset for the NAFLD and hypertension studies is not provided with the paper or supplemental material

Reviewer #2: Yes

PLOS authors have the option to publish the peer review history of their article (what does this mean?). If published, this will include your full peer review and any attached files.

Reviewer #1: No

Reviewer #2: No
---

## [Decision Letter · Decision Letter 1]

22 Jan 2021

Dear Prof. Dr. Kaderali,

We are pleased to inform you that your manuscript 'From heterogeneous healthcare data to disease-specific biomarker networks: a hierarchical Bayesian network approach' has been provisionally accepted for publication in PLOS Computational Biology.

Best regards,

Alison Marsden

Associate Editor

PLOS Computational Biology

Florian Markowetz

Deputy Editor

PLOS Computational Biology

Reviewer's Responses to Questions

**Comments to the Authors:**

Reviewer #1: The authors have addressed all my comments. Publication of this contribution in its current form is therefore recommended.

Reviewer #2: I believe that this revised version is improved and adequately addresses the points raised by the reviewers.

**Have all data underlying the figures and results presented in the manuscript been provided?**

Reviewer #1: Yes

Reviewer #2: Yes

PLOS authors have the option to publish the peer review history of their article (what does this mean?). If published, this will include your full peer review and any attached files.

Reviewer #1: No

Reviewer #2: No

---

## [Editor Report · Acceptance letter]

7 Feb 2021

PCOMPBIOL-D-20-01495R1 

From heterogeneous healthcare data to disease-specific biomarker networks: a hierarchical Bayesian network approach

Dear Dr Kaderali,

I am pleased to inform you that your manuscript has been formally accepted for publication in PLOS Computational Biology. Your manuscript is now with our production department and you will be notified of the publication date in due course.

With kind regards,

Alice Ellingham
